# Muscle Mechanics and Thick Filament Activation: An Emerging Two-Way Interaction for the Vertebrate Striated Muscle Fine Regulation

**DOI:** 10.3390/ijms24076265

**Published:** 2023-03-27

**Authors:** Lorenzo Marcucci

**Affiliations:** 1Department of Biomedical Sciences, University of Padova, 35131 Padova, Italy; lorenzo.marcucci@unipd.it; 2Center for Biosystems Dynamics Research, RIKEN, Suita 565-0874, Japan

**Keywords:** mechanosensing mechanism, thick filament regulation, SRX, dual filament activation

## Abstract

Contraction in striated muscle is classically described as regulated by calcium-mediated structural changes in the actin-containing thin filaments, which release the binding sites for the interaction with myosin motors to produce force. In this view, myosin motors, arranged in the thick filaments, are basically always ready to interact with the thin filaments, which ultimately regulate the contraction. However, a new “dual-filament” activation paradigm is emerging, where both filaments must be activated to generate force. Growing evidence from the literature shows that the thick filament activation has a role on the striated muscle fine regulation, and its impairment is associated with severe pathologies. This review is focused on the proposed mechanical feedback that activates the inactive motors depending on the level of tension generated by the active ones, the so-called mechanosensing mechanism. Since the main muscle function is to generate mechanical work, the implications on muscle mechanics will be highlighted, showing: (i) how non-mechanical modulation of the thick filament activation influences the contraction, (ii) how the contraction influences the activation of the thick filament and (iii) how muscle, through the mechanical modulation of the thick filament activation, can regulate its own mechanics. This description highlights the crucial role of the emerging bi-directional feedback on muscle mechanical performance.

## 1. Introduction

The most important function of vertebrate striated muscles, cardiac and skeletal, is to generate force and motion to produce mechanical work. Because the same contracting structure must serve different purposes, such as locomotion and posture, breathing or supplying blood to every organ, it requires a flexible and fine regulation. Recent groundbreaking discoveries in this field are the topic of this review.

Muscle contraction is generated at the molecular level by the ability of myosin motors to convert chemical energy of ATP into mechanical work. In the cylindrical 2 µm-long structural units of the striated muscle, called sarcomere, the 150 nm-long tails of the myosin molecules are arranged to form a thick filament, from which two 20 nm-long heads per dimer protrude in a quasi-helical arrangement (see Figure 1). Each head (the myosin globular Subfragment 1, S1) is formed by a catalytic domain and a lever arm, which hosts two light chains, the regulatory light chain (RLC) and the essential light chain (ELC). The thick filaments run symmetrically around the center of the sarcomere, the M-line, and are attached to its two extremities, the Z-lines, through the giant protein titin, which can transmit passive forces to the thick filament to prevent excessive elongations of the passive sarcomere length (SL). From the Z-lines originate, with opposite polarity, the thin, actin-based filaments, which run parallel to the thick filaments. The catalytic domain can attach actin monomers while the lever-arm performs the so-called power stroke, generating force and relative sliding of the filaments (Figure 1, upper part). After the power stroke, the nucleotide-binding pocket of the myosin head has an open configuration. When the myosin head rebinds ATP, which allows the detachment from actin, it assumes a pre-power- stroke state, characterized by a closed structure of the pocket. The symmetric structure makes the half-sarcomere the smallest contractile unit in muscle [1].

The classical description of the force regulation in striated muscle is based on the thin filament-mediated calcium signaling pathway. When the free calcium concentration ([Ca^2+^]) is low, the actomyosin interaction is inhibited by the regulatory proteins on the thin filaments which hide the myosin binding sites (a situation depicted by the brown energy barrier in the lower part of Figure 1). When [Ca^2+^] increases, released by the sarcoplasmic reticulum following an action potential, these regulatory proteins undergo to a structural change, releasing the myosin binding sites and “activating” the thin filaments [2,3]. Then, during relaxation, the calcium is taken up by the sarcoplasmic reticulum, and the thin filaments are inhibited again. In this classical description, other proteins are supposed to play a role in the activation process mainly influencing the calcium sensitivity of the thin filament. The myosin binding protein C (MyBP-C), a protein closely associated with the myosin motor in the central third of the thick filament (C-zone, see Figure 1, left part of the thick filament) in a ratio of 1:6 with myosin, is known to interact both with myosin and actin [4,5]. The myosin molecules itself, when attached to actin, stabilize the active conformation of the thin filament regulatory proteins [6]. Nonetheless, in this calcium-mediated thin filament activation paradigm, myosin motors are basically immediately ready to interact with actin monomers when available, so it is not surprising that their characterization in relaxed muscle has received much less attention than during activation.

Over the last decade, the interest in the myosin stable states in the relaxed muscle has greatly increased, prompted by the experimental evidence that myosin motors can stay in an autoinhibited state even when actin filaments are activated. Thus, the thin filament regulation paradigm must be extended in the view that muscle contraction requires the activation of both the thin and the thick filaments. While historical and more recent experimental results have been extensively summarized in several reviews [7,8,9,10,11,12,13,14,15], here, special attention is paid to the mechanical implication of the newly discovered thick filament activation. The most relevant techniques which probe the activation level of the thick filament, and its known modulators, will be introduced. Then, the experimental evidence (even within each work) will be divided into three categories. First, how external modulations of the thick filament activation influence muscle mechanics, and then, how external mechanical interventions affect the activation of the thick filament, showing bi-directional feedback, which will introduce the third category of evidence: how the modulation of the thick filament activation, induced through mechanical interventions, affects the muscle mechanical response. Importantly, this represents an internal feedback system readily available to the muscle to regulate its contraction to the required task. This description can help to better appreciate the crucial role of the thick filament regulation on the vertebrate striated muscle mechanical performance.

## 2. Detached Myosin Stable States

The two heads in each myosin dimer, responsible for the ATP hydrolyzation and the force generation, are subjected to thermal (random) fluctuations and can oscillate between stable states (Figure 1, lower part) [16], i.e., states that are populated long enough to be observed through some available technique. At present, the techniques of main interest for this review, which allow us to differentiate the stable states assumed by the myosin molecules when detached from actin, can be summarized as follows:

### 2.1. X-ray

Thanks to the highly ordered structure of the sarcomeres, time-resolved X-ray diffractions can be exploited to characterize the relative position of the myosin heads in the relaxed muscle and the changes during activation, in physiological condition [17]. In vertebrate muscle, myosin motors are distributed along the thick filament forming 49 layers of three dimers each, with an axial spacing of about 14.5 nm, which repeat themselves every three layers, generating a fundamental axial periodicity of about 43 nm. This forms a three-stranded helical arrangement, which produces a series of “myosin layer-line” (MLL) reflections, and axial reflections. The X-ray reflections that are assumed as signatures of the activation of the thick filament come from the helical order of the myosin heads, mainly MLL1 and, as proposed in mechanosensing mechanism [18] (see below), from the short axial periodicity, mainly the spacing of M3 and M6 (S_M3_ and S_M6_), which increases of about 1% during muscle activation.

More recent developments facilitate milliseconds time-resolved analysis and, under some assumptions, also for a localization of the structural changes in the myosin motors along the thick filament [18,19,20]. However, even early X-ray analyses were able to characterize a quasi-helical order of the myosin heads in the relaxed muscle and its loss during activation [17,21,22]. Notably, as more extensively reviewed in [12], since these early observations, Haselgrove [23] advanced the hypothesis that the loss of order could be evidence of a myosin-based regulation and not only a consequence of the interaction between myosin and actin, but the rising actin-based paradigm needed more evidence to be questioned.

### 2.2. CryoEM

A structural support to the well-organized quasi-helical conformation of the myosin heads on the thick filament in relaxed muscle comes from the electron microscopy both for the whole thick filament and for the sub-fragments of the myosin molecule (for a historical review see [24]).

In the tarantula relaxed thick filament [25], the two heads in the myosin dimers were shown to form the so-called Interacting Heads Motif (IHM, see Figure 2, left panel) through a head–head and head–tail intra-molecular interaction, as originally observed in a smooth muscle myosin [26]. Consecutive IHM can be packed along the backbone in a helical lattice, through inter-molecular interactions between the neighbor molecules. IHM plays an important role in the mechanism of activation of the thick filament [27]. In the relaxed muscle the dimer in the IHM configuration has an asymmetric structure with a “blocked head” lying on the thick filament and docked to its S2 in a pre-power stroke conformation, which requires the closed structure of the pocket of the myosin head and prevents both actin interactions and the ATP hydrolyzation. The second myosin of the dimer, the “free head”, can sway hydrolyzing ATP and eventually interact with actin. However, it can also dock onto the blocked head, forming the autoinhibited state (Figure 2, right panel).

It is important to note that differently from vertebrate striated muscle, tarantula skeletal muscles have a myosin-based regulation, driven by phosphorylation of the RLC, which destabilizes this packed conformation, releasing the heads for the contraction. Moreover, it has a four-fold symmetry, and the observed structures may be not straightforwardly transferred into the three-fold symmetric vertebrate muscle. However, the packed conformation has also been confirmed in scallops, where the myosin motors are directly regulated by calcium binding to the ELC, and later in isolated thick filaments from vertebrate cardiac muscle [28,29]. Despite a recent work challenging the interpretation of the X-ray diffraction in relaxed muscle through the existence of the IHM configuration, at least in bony fish muscle [30], a subsequent, more detailed modelling analysis confirmed it [31]. Another recent work confirmed that the X-ray diffraction patterns in tarantula skeletal muscle can be well explained using a structure based on the IHM [32]. The presence of this structure in different muscle types and species, separated by millions of years of evolution, suggests its fundamental role in the fine regulation of muscle contraction [33].

High-resolution reconstructions of purified muscle heavy meromyosin (HMM) consisting of the two heads (S1) and of the initial portion of the tail (S2, see Figure 1) also showed the intra-molecular interactions which lead to the inhibited state, showing that the whole filament structure is not needed to form it [10 and ref. therein].

Importantly for what will be said later, despite the resolution limits, EM structural analysis of the HMM suggests that the S1 in the sequestered state interacts with the S2 through a flat region, called Mesa [34], as also supported by a recent 2.33 Å resolution structure of the Bovine Cardiac Myosin S1 [35].

### 2.3. Mant-ATP

Incubating permeabilized fibers in fluorescent nucleotide analogs of ATP, followed by a chase with ATP, allows the measurement of the ATP turnover rate of myosin. During the chase phase, after rapid exchange from the solution containing mant-ATP to a relaxing solution with a high concentration of ATP, the fiber remains relaxed, and the fluorescence intensity decreases with the release of the mant-ATP. A multi-exponential fitting of the time-course of their fluorescence decay showed that in relaxed muscle fibers, myosin motors can be observed at least in two biochemically stable states [36]. One had an ATP turnover rate in the order of magnitude of 0.01 s^−1^, as also observed in purified proteins in in vitro experiments in the absence of actin, which is two orders of magnitude slower than during the actomyosin interaction (order of magnitude 1 s^−1^). The second subpopulation of detached myosin had an even slower ATP turnover rate, in the order of magnitude of 0.001 s^−1^ per myosin head. The latter biochemical state has been defined super-relaxed state (SRX) in contrast the disordered relaxed state (DRX) with higher ATP turnover rate. SRX was observed initially in rabbit skeletal [36] and cardiac [37] relaxed muscle. The SRX signal disappeared in partially activated skeletal muscle (about half of the maximum tension) but not in the cardiac muscle [37].

More recently, the technique was also applied to human skeletal muscle fibers [38], showing a fiber-dependent distribution between the two states. It was also applied to myofibrils and associated with rapid mixing methods [39], showing that, from the rigor state, the SRX state can be repopulated in as fast as 200 milliseconds.

Since its discovery, SRX has been understood in relation to the ordered structural states described before [36] (Figure 2). ATPase activity is sterically inhibited in the pre-power stroke configuration required for the blocked head, leading to a very low ATPase rate. The free myosin head is less constrained and has been proposed to sway, intermittently forming the DRX state [27]. However, the SRX as a biochemical state was also shown in purified bovine and human cardiac myosin both with a shorter long tail (two-headed with the first two heptads of proximal S2, 2 hep HMM) and also in its absence (single-headed with no S2, sS1) [40,41], even though in a smaller proportion than in the two- headed with the first 25 heptads of proximal S2 (25 hep HMM). In the shorter and single-headed constructs the intra-molecular interactions, seen in the IHM, cannot be formed (for a recent review on this subject see [11], see also below).

### 2.4. RLC-Probes

The conformation of the myosin heads in demembranated fibers can also be analyzed using fluorescent probes attached at two different lobes, N and C, of the RLC [42]. The order parameter “P2”, obtained by fluorescence polarization, can be used to estimate how parallel the probe dipole is to the thick filament axis, and therefore to estimate the relative distribution of the motors between the packaged and the disordered states (schematically represented in Figure 2, right panel). Importantly, the method can provide a time-resolved structural transition on the millisecond timescale, and, as with the force-pCa curve, the P2-pCa curve can be fitted by a Hill equation [43], informing on the cooperativity of the thick filament activation.

The detached stable states probed by these different techniques are not necessarily coincident. Several studies in the last few years clarified their reciprocal relationships as reported in the next section.

## 3. Non-Mechanical Modulators of the Detached Myosin Stable States and Reciprocal Relationships

Every event able to perturb the stable states observed with the above-mentioned techniques represents an important tool not only to test their influence on muscle mechanical tasks, but also to verify whether they are each other related or even representative of the same myosin conformation.

### 3.1. Temperature and Ionic Strength

Among the first evidence used to probe a relationship between SRX and helical order was the effect of the temperature which decreased the SRX population [36] similarly to the decrease in ordered myosin heads seen by X-ray diffraction [44,45,46,47]. RLC-probe experiments also showed a sigmoidal dependence of the P2 parameter with the temperature, indicating a higher population of motors close to the thick filament at physiological temperatures than at lower temperatures both in rabbit skeletal muscle [42] and rat cardiac trabecula [48]. Proposed reasons for the stabilization of the packed configuration with the temperature are that it needs the pre-power stroke configuration of the myosin motor, which is stabilized at higher temperatures [49], and that due to the hydrophobic effect, the higher temperatures stabilize several protein–protein interactions [50].

The temperature effect must be taken in proper consideration when comparing different experimental evidence. In mouse skeletal muscle, lowering the temperature reduced the motors with a packed conformation in the relaxed intact fibers but also made them unavailable for the actomyosin cycle during activation, putting them in a refractory state which can be related to the hibernating mechanism [51]. Two recent works pointed out the crucial importance of controlling the temperature and the lattice spacing to fully understand the role of the thick filament activation on the muscle contraction, indicating that at least 25–26 °C is needed to reach quasi-physiological conditions both for skeletal skinned [49] and cardiac demembranated [48] muscle in vertebrates.

Similarly to the temperature, higher ionic strengths destabilize the SRX state in both synthetic thick filament [52] and in the 25 hep HMM, but not in the 2 hep HMM or sS1 [40,41]. These observations are compatible with the idea that the biochemically defined, low ATPase turnover rate, SRX state is stabilized by charge–charge interactions in the presence of the long tail, but not in its absence, showing that it is tightly related, but not coincident, to the structural state of IHM.

### 3.2. Small Molecules Directly Targeting the Myosin Motors

There is an emerging class of pharmaceutical agents which directly target the sarcomeric muscle myosins [13], and several compounds have been identified to directly modulate the stable states of myosin when detached from actin, providing an important source of information about the impact of these states also in the activated muscle. Piperine was the first modulator identified, which specifically affects the skeletal muscle SRX, with possible applications for the treatment of obesity and diabetes through the alteration of the metabolic rate at rest [53]. For cardiac muscle (but also to a lower extent for skeletal muscle, see later), a very well-studied inhibitor of myosin motors is the MyoKardia’s mavacamten, currently an FDA-approved treatment, commercialized under the name of camzyos™, that targets the source of symptomatic obstructive Hypertrophic CardioMyopathy (HCM). Discovered as an ATPase inhibitor which acts on the Pi release [54], mavacamten stabilized the SRX state in 25 hep HMM human beta-cardiac myosin to almost 100% of the motors [41], with a 20-fold reduction in the basal ATPase rate in the conditions used in that work. Further experiments showed a dose-dependent reduction in ATPase rate in the bovine cardiac muscle, and even at saturating concentration of mavacamten, the presence of actin disrupted the autoinhibited state [40].

A big step in the comparison between the stable states of detached myosin, described in the previous section, was made with the electron microscopy analysis on the human cardiac 25 hep HMM, which showed a stabilization of the folded-back, or closed, heads configuration with mavacamten quantitatively compatible with the ATPase data [41]. Moreover, ATPase turnover rate and X-ray signatures of thick filament activation showed similar inhibition when both porcine and human cardiac muscle fibers were treated with mavacamten [41].

Despite the evident relationship between SRX and IHM, mavacamten confirmed that the two states cannot be fully superposed. Mavacamten stabilized, despite doing so to a lower extent, the SRX state in 2 hep HMM and in sS1 (lowering the energy in Figure 1), with no significant difference between the two cases, supporting the idea that the S2 is needed to stabilize the SRX but not to create it [41]. Additionally, in purified bovine cardiac HMM myosin, despite the fractions of myosin in the IHM (forming the intra-molecular interactions as probed by time-resolved fluorescence resonance energy transfer) and SRX state in solution being similar, the treatment with mavacamten increased the SRX state by a much greater extent than the increase in the IHM state (55% vs. 4%, respectively), confirming that the IHM is “sufficient but not necessary” for the SRX state [55]. Notably, mavacamten may reduce the ATPase rate stabilizing the pre-power stroke state in the myosin motors, structurally favoring the formation of the SRX state [10,50].

The development of compounds that directly target the myosin motor to treat cardiomyopathies is very promising, and different drugs are currently at different clinical trial stages [14]; however, despite the fact that mavacamten also affects other steps of the actomyosin cycle [40,56,57], the extensive study made on it in the last five years represents a unique source of information to understand the relationship between the stability of the SRX state in relaxed muscle, and their performance when activated, as shown below.

### 3.3. RLC and MyBP-C Phosphorylation

The ways of modulation of the thick filament activation described above are external interventions, useful to highlight indirectly the structural basis of the thick filament regulation but not within the reach of muscle in vivo. While the physiological signals that disrupt the inter- and intra- molecular interactions, allowing the myosin heads to become available to the force generation cycle, are still poorly understood, a clear role has been attributed to the phosphorylation of both the RLC and MyBP-C.

The RLC phosphorylation was shown to regulate the packed configuration in cardiac muscle through RLC-probes [58,59], X-ray analyses [60] and SRX estimation in skeletal muscle [36], and in human cardiac 25 hep HMM but not in the 2 hep HMM [61]. Physiologically, it has a role during repeated stimulation, while it may be relatively too slow to impact a single twitch [62]. Notably, in relaxed demembranated cardiac muscle, the role of the RLC phosphorylation on the activation of the thick filament was shown to be temperature dependent, showing minor effects at quasi-physiological values [48].

Several studies (reviewed in [8]) showed that MyBP-C stabilizes the autoinhibited state of the myosin, and that its phosphorylation has a regulatory role on the activation of the thick filament [63,64,65]. Coherently, spatially resolved ATPase rates along the thick filament showed a higher proportion of SRX-nucleotide-binding events in the C-zone [66]. In interpreting the effects of MyBP-C on the force production, it is important to remember, however, that it can bind both myosin and actin, increasing the calcium sensitivity of the latter, assuming a double role of activator and inhibitor of the muscle contraction [63]. Additionally, besides RLC and MyBP-C, several other proteins can be phosphorylated in striated muscle, both at the myofilament level and associated with the excitation–contraction coupling. Although they do not seem to be related to the thick filament regulation, this may influence some experimental observations.

### 3.4. Genetic Modifications or Alterations

As said, the flat region on the myosin head called Mesa was indicated from EM analyses as crucial for the stabilization of the autoinhibited state of the motor, and was shown to be a hotspot for several genetic mutations leading to hypertrophic cardiomyopathy (HCM) [10]. Several of them were proved to alter the stability of the HMM autoinhibited state [40,56,61,67]. HCM variants in left ventricle myocardium from mice and in cardiomyocytes derived from human-induced pluripotent stem cells (iPSC-CMs) showed a reduced proportion of SRX [68]. Two HCM mutations (R403Q and R663H) with moderate effects on single molecule contractility parameters, destabilized the SRX state in both single HMM [69] and single fibers [41]. The same effect was shown on skinned papillary muscles of mice carrying mutations in the human ventricular RLC [70]. Moreover, not only do many HCM mutations occur in the myosin Mesa, potentially interfering with the protein–protein interactions which generate the sequestered state, but many others also occur in the MyBP-C, which is also involved in SRX stability [71]. In particular, while MyBP-C stabilized the SRX state in WT HMM, this ability was lost in the presence of the said two mutations [69]. Comparing the performance of contraction in WT muscle and muscle carrying the HCM-related genetic mutations is then informative to understand the role of the thick filament regulation in this pathology.

## 4. Thick Filament Activation and Contractile Performance

The evidence reported in the previous sections has shown the existence of more than one stable states, partially overlapping or at least related each other, of the myosin head when it is not attached to actin, the configuration required to generate force. Therefore, the existence of these states does not imply per se that they are important for muscle contraction. Clearly, the packed configuration represents a geometrical hinderance for the actomyosin interaction and must be abandoned to generate force. Similarly, the SRX state has a limited, or even completely prevented [39], ability to hydrolyze ATP, the step needed to supply energy to produce work. However, even in the actomyosin cycle, several stable states, despite being widely accepted, can be neglected in mathematical models because their probability of occupancy is relatively low, and, as an extreme example, even the two-states (attached–detached) model originally proposed by Huxley [72] is sufficient to describe a variety of muscle properties. As mentioned before, the packed configuration and its loss during activation was observed in X-ray studies several decades ago, but only recently has its role in muscle regulation been gaining interest. What is the experimental evidence which prompted this?

The fact that the myosin Mesa is both crucial for the stability of the autoinhibited state and hotspot for several HCM genetic mutations suggested the intriguing hypothesis, proposed by James Spudich, that some genetic mutations induce HCM not modifying the actomyosin complex mechanical properties but modifying the number of motors (Na) which are available to generate force [8,10]. This “mesa hypothesis” postulates that these mutations weaken the sequestered state, increasing Na and causing hypercontractility, which leads to hypertrophy and the subsequent clinical manifestations, thus suggesting the crucial role of the myosin-based regulation of the contraction, at least in cardiac muscle. Then, the emerging picture introduces Na as a new factor in the determination of the force generated by an ensemble of sarcomeres, besides the three classic parameters: the level of activation of the thin filament, the intrinsic force of each myosin and its duty ratio [34]. This implies a shift in the paradigm, summarized in [12], from a thin filament-based activation, to a “dual filament activation”, where the regulation of the thick filament, missing a regulatory protein as in the thin filament, relies on the modulation of Na under different external conditions.

In what follows, stretching a little the nomenclature proposed in [18], and under the above-mentioned limits, the detached myosin available for contraction will be defined as in an “ON” state, while all the other stable states will be generally included in an “OFF” state, where the motors are not ready to interact with actin even when the thin filament is activated. The relationship between the stable states of detached myosin motors and muscle mechanical response represents a highly active topic of discussion and the mechanism which modulates the OFF–ON transition in vivo is not fully understood. The data about this relationship will be discussed in the remaining part of this work, following the three categories mentioned in the introduction: the influence of the external regulation of the thick filament on muscle contraction, the influence of the external mechanical interventions on thick filament regulation, and the bi-directional feedback. Thick filament activation is only one piece of the muscle highly complex regulatory system, and sometimes it is not straightforward to follow these three categories, but this approach will help to keep a focus on the muscle mechanics.

## 5. From the Activation of the Thick Filament to Muscle Mechanical Response

The external interventions which can modulate the OFF–ON transition described above, despite being limited in highlighting its physiological regulation, are informative about its influence on the force generation.

The effects of the temperature on muscle contraction are not straightforward to be interpreted in terms of OFF–ON equilibrium because temperature also strongly affects the kinetics of the actomyosin cycle and the force per motor, as also recently shown in [73]. However, in the same study [73], a known effect of the temperature on the force-pCa curve has been associated with the thick filament regulation. The force-pCa curve is an example of data that cannot be easily included into one of the three above-mentioned categories. It is obtained primarily modulating the thin filament activation at different constant calcium concentrations. The higher is the [Ca^2+^], the more actins there are available to the myosin, and the higher is the force. In this sense, it could be seen as an external mechanical intervention. However, the higher levels of thick filament activation at higher forces, shown by all the testing methods, could be interpreted just as a passive adaptation, if not associated with a secondary perturbation. The nature of the latter will be used here below to categorize the experiment. The force-pCa curve has a sigmoidal shape and is characterized mainly by three parameters: the maximum tension T_0_ at saturating [Ca^2+^], the [Ca^2+^] at which the force is T_0_/2, or its logarithm pCa50, and the steepness of the sigmoidal curve, called the Hill parameter n_H_, from the Hill’s equation used to fit the data points. n_H_ is an index of the cooperativity of the muscle activation and is known to be different (i.e., a careful fitting requires more than one Hill’s curve) at low levels and at high levels of calcium activation at physiological temperatures. Since in skeletal muscle the SRX state is completely destroyed at half activation [74] (however, see also below), this difference may be an evidence of the cooperativity induced by the thick filament. In support of this, the asymmetry disappeared not only at low temperatures (12 °C) but also at high temperatures (35 °C) when a temperature jump technique is used, which requires the activation of the fibers starting from an activating solution in equilibrium at 1 °C, where the OFF state can be supposed to be almost destroyed [73]. It is also worth noting here that both the RLC probe order parameter P2 and the force have a sigmoidal dependence on pCa, but P2 showed a higher n_H_ and pCa50 than force both in skeletal [43] and cardiac [59,75] muscle. The different cooperativity indicates that the activation of the thick filament is not passively following the activation of the thin filament, and most likely, it has an important role in the steepness of the force-pCa itself. The P2 high cooperativity is possibly derived by an axially and laterally propagation of a change triggered on one region of the long myosin tail within the backbone, as suggested by the atomic structure of the complete myosin tail from flight muscle [76]. A single structural change could alter the structure of six myosin crowns. Additionally, a not uniform change might be more consistent with the short axial periodicity intensity change.

The effects of RLC phosphorylation were extensively reviewed in [7]. Phosphorylating the RLC of vertebrate cardiac muscle by the myosin light chain kinase (MLCK), increased resting and maximal forces, lowered pCa50 and increased the force development rate. Similarly, transgenic mice, where the RLC phosphorylation was prevented, showed a reduction in isometric force and maximal power. The effects of phosphorylation can be associated with the negative charge of the bound phosphate group, which might destabilize the OFF state ([7] and ref. therein). Recently, the influence of RLC phosphorylation-mediated thick filament activation on the cardiac force-pCa relation was specifically addressed [59]. Probes attached to either WT cardiac RLC (cRLC) or a modified cRLC which cannot be phosphorylated, showed a similar increase in pCa50 after cMLCK treatment, despite, in the latter case, only the unlabeled motors being phosphorylated, indicating that the activation effect is transmitted through a cooperative thick filament structural change [59].

Phosphorylation of the MyBP-C increased the power output in skinned cardiac myocytes in rat [77] and mice [65], while the increase in calcium sensitivity, due to protein kinase A (PKA)-mediated phosphorylation of myofilament proteins, was blunted in hearts expressing non-phosphorylatable cMyBP-C, compared to WT [78]. Adding the β-adrenergic effector isoprenaline to induce MyBP-C phosphorylation in rat intact cardiac muscle doubled the peak force, even without affecting the regulatory state in the thick filament in relaxing condition [79], an observation which will also be discussed below. In slow skeletal muscle, a treatment with PKA doubled the force overshoot in a slack–restretch protocol, but only at low [Ca^2+^], a difference that the authors associated with the different activation of the thick filament [80]. Similar effects, again at submaximal calcium activation, were shown on rat myocardial preparations through a slack–restretch maneuver. Additionally, modifications which decreased the SRX population increased the rate of force redevelopment [65].

As an extensively studied stabilizer of the OFF state, mavacamten represents a precious source of information to understand the effects of the thick filament regulation on the muscle contraction. The induced reduction in the ATPase activity was associated with a reduction in the maximum force and fractional shortening in skinned rat cardiac myofibrils and, in vivo, to a reduction in fractional shortening in both WT and HCM mutant mice [54]. Coherently to the stabilization of the OFF state in the HMM, mavacamten reduced the sliding velocity of an actin filaments propelled on a bed of cardiac HMM [56]. The decrease in the maximum force and shortening velocity after treatment with mavacamten was shown also in iPSC-derived cardiomyocytes from human engineered heart tissue [81]. More recently the mavacamten-associated lowering of maximal tension was observed also in both permeabilized porcine fibers from ventricular myocardium at 22 °C [82] and permeabilized human myocardium at the physiological temperature of 37 °C [57]. It is worth noting that pCa50 and n_H_ are not affected by mavacamten in the former work [82], while pCa50 is decreased and n_H_ increased by mavacamten in the latter [57]. Comparing human ventricular and rabbit psoas myofibrils, both muscle types showed a dose-dependent decrease in maximal tension when exposed to mavacamten, confirming that it has an effect also on skeletal muscle, despite having a sensitivity one order of magnitude smaller than in human ventricular myofibrils at 15 °C [83].

In the same work, mavacamten was used also to test the effects of thick filament regulation on the kinetics of force generation. Myofibrils were subject to sudden solution changes by rapidly translating them between two flowing streams, to overcome diffusional barriers. The effect of the drugs was extremely fast and fully reversible. Increasing doses of mavacamten reduced the kinetics of force generation, and regeneration after a fast release–restretch protocol, in fast skeletal muscle. However, it had no (or even contrary at high doses) effects on the rate of force generation in slow cardiac muscle at low temperatures [83]. This disagreed with the effect of mavacamten on the rate of force generation observed at higher temperatures in mice trabecula [84] and skinned human ventricular strips [57] (at 25 °C and 37 °C, respectively). The discrepancy could be related to the temperature dependence of the Pi release step. Interestingly, mavacamten had no significant effect on force relaxation on fast skeletal muscles, while it made relaxation faster in cardiac muscle [83]. Despite the discrepancies, these works clearly showed a role of the thick filament regulation on the kinetic of force generation and relaxation.

A direct interpretation of the mechanical consequences of altered OFF–ON regulation due to myosin mutations is made difficult by possible chronic adaptations. However, when associated with the effects of the FDA-approved treatment of HCM, mavacamten, clear evidence emerges. Functional analyses on mice cardiac muscle and human iPSC-CMs with HCM variants associated with a reduced number of motors in the SRX state, thus with higher Na, showed enhanced contractility and impaired relaxation, while application of mavacamten restored those values toward the WT case [68]. Additionally, application of mavacamten regulated the OFF–ON equilibrium and the force in HCM mutations related to MyBP-C in mice [71,84]. Importantly, in a clinical study of more than 4500 patients, among those who had rates of heart failure and atrial fibrillation intermediate between pathogenic and benign variants, the higher rates were shown by the group whose variants were associated with the formation of IHM [68]. In the first work presenting mavacamten [54], there was already evidence that its chronic administration to mice with HCM-related mutations, helped in attenuating the disease development. Since then, several studies on in vivo effects were reported (for recent reviews see [13,14]), including the clinical trials [85,86] which led to the FDA approval. These probably represent the strongest evidence that the regulation of the detached myosin stable states deeply affects the physiology of contracting muscle.

## 6. From the Mechanical Perturbation to the Thick Filament Activation

A milestone discovery in the role of the thick filament activation on the muscle contraction was that not only does a modulation of the OFF–ON equilibrium alter the muscle contraction, but the opposite is also true. After a stimulus, the thin filament reaches the full activation in few ms, while the active force, and thus the activation of the thick filament, starts after a “latent period” of about 10 ms [87]. Linari and co-workers associated the time-resolved measure of the X-ray signatures of the thick filament activation with the control of the force on a frog intact fiber switching from isometric conditions to unloaded isotonic contraction [18]. In isometric conditions, the OFF-state X-ray signatures were partially lost at a very low tension of 0.1 T_0_, but imposing a rapid shortening (5% of fiber length for 20 ms) to keep the filaments at zero tension at the end of the latent period was sufficient to preserve the OFF state. This was the first evidence of the “mechanosensing mechanism” which postulate that the activation of the thick filament is physiologically regulated by the tension acting on it (in red in Figure 1). To further test this idea, the authors dropped the force to zero, imposing the fast shortening (10% L0 for 40 ms) at the fully activated muscle at T_0_, showing that this induced the thick filament to be partially switched OFF. Despite being rapid, this mechanically induced switching OFF process had a half-time of about 20 ms, much slower that the half-time of decrease in force (about 3 ms), indicating that the process was limited by a transition in the detached heads. In the proposed paradigm, the existence of few “constitutively ON” motors is sufficient to sense the activation of the thin filament and activate other heads, as well as to sustain alone the rapid shortening during unloaded contraction [88], suggesting the existence of an internal feedback in muscle regulation, which can modulate the available motors depending on the external conditions. It is worth noting that one of the X-ray signals of the activation for the thick filament, as proposed by the mechanosensing mechanism at least in the fast-twitch muscles, is an increase in the backbone axial periodicity much higher than its compliance, which has been associated with the disruption of the inter- and intra-molecular connections which are needed to form the IHM structure and the helical arrangement [9], a conclusion supported also by structural studies on insect muscle [89].

The mechano-sensing mechanism was shown to sense not only the active tension but also the passive tension in an analysis of the RLC probes in passively stretched demembranated single fiber of rabbit psoas (pCa 9, 25 °C) [43]. Stretching the fibers from 2.40 µm to 3.60 µm increased the passive tension, mainly due to titin, from zero up to 200 pN per thick filament, about 1/3 its T_0_. The change in the parameter P2 with the tension indicated that also the passive tension can activate the thick filament. Fast steps of 110 pN per thick filament, kept constant during the viscous length response by a feedback system, showed that the activation and the deactivation is not instantaneous but has an intrinsic rate constant of about 250–300 s^−1^. Moreover, using blebbistatin to prevent contraction, it was shown that the calcium concentration has negligible effects on the P2 at different passive forces. P2 showed a linear relationship in the range of passive force of 20–300 pN per thick filament. Strikingly, analyzing the P2 parameter during activation at different level of calcium in the absence of blebbistatin, the data showed that passive and active forces have almost the same effect on the thick filament activation.

It is worth noting that, besides representing an independent support to the mechanosensing mechanism and showing that it is also sensitive to passive tension, the approach adopted in this study [43] further limited the role of the activated actin in determining the equilibrium between the OFF and ON states of the myosin motors. In fact, the equilibrium between ON motors, attached and detached, indirectly alters the equilibrium between the detached ON and OFF motors. Imposing a shortening at maximum velocity quickly detaches the myosin motors at a half-time of about 3 ms [18], bringing the attached vs. detached ON equilibrium to a situation close to that in relaxed muscle, not only dropping the tension but also increasing the detached ON, and indirectly the OFF, population, blurring the sole effect of the force. Passively stretching the sarcomere at low pCa overcomes this limit. This is coherent also with earlier evidence that myosin motors in active muscle was unfolded also in the non-overlapping zone, in the absence of actin filament [90].

Importantly, the mechano-sensing mechanism activate the thick filament in subsequent steps, as showed with a high resolution time-resolved X-ray analysis on intact extensor digitorum longus (EDL) muscles of mice electrically stimulated at near physiological temperature of 28 °C [91,92]. First the helical order of the myosin motors was lost (half-time 8 ms), with the synchronous change in the spacing of the filament backbone, and only later the motors were released from the folded conformation on the filament backbone [91]. At rest, as at the plateau of the isometric contraction, there was no inhomogeneity in the distribution of the ordered motors along the thick filament. However, during the imposed shortening after the latent period and immediately after it, the disordered motors appeared to be concentrated near the tip of the thick filament, suggesting different roles of its zones in the muscle activation [92].

The importance of the role of the fine regulation of the thick filament in the skeletal muscle contraction was confirmed by X-ray analysis of a single twitch, which showed that, despite the thick filament being confirmed to be highly activated above 0.5 T_0_ (but possibly not fully in mammalian muscle, see also [93]) in the tetanic stimulation, the lower tension at the peak of the twitch (0.4 T_0_) was limited by the partial activation of the thick filament itself, besides the transiently activation of the thin filament [91].

The previous data showed the importance of the mechano-sensing mechanism in skeletal muscle; however, it is also present, possibly with a more crucial role, in cardiac muscle. This was first shown by X-ray diffraction patterns in intact ventricular trabeculae from the rat at rest and at the peak force of a twitch in two different conditions: when the SL was allowed to reduce against the compliance of the attachments in a fixed end (FE) protocol and when it was kept at a longer fixed length (FL) through a feedback mechanism [94]. While resting SL was 2.2 µm in both cases, SL and force at peak (Tp) were 1.9 µm and 44 kPa for FE, and, respectively, 2.1 µm and 93 kPa for FL. Through a structural model of the sarcomere, the diffraction patterns were used to deduce the number of motors in the different configurations at rest and at Tp. Interestingly, in view of the role of the thick filament regulation on the active contraction, in this intact cell preparation, most of the motors were in the OFF state at rest, and they were activated by the increase in tension after the stimulus [94]. Importantly, the population in the ON state was proportional to Tp, also implying a constant duty ratio in the two protocols, which is consistent with the fact that both protocols lead to isometric conditions. In support of this conclusion, a recent mechanical test in the same preparation provided direct evidence that Tp is directly related to the number of attached motors with an average force which is independent of Tp itself [95]. In other words, Tp was regulated by Na, the number of motors available for contraction, and thus by the activation of the thick filament. Importantly, in these two protocols, Tp regulation was not mediated via a different SL in the relaxed muscle, a mirror of the diastolic conditions of the heart, but via a different external load against which the contraction occurred during a single twitch, mimicking a high (LC) or low (FE) aortic pressure.

A sequential activation of the thick filament was also shown at a quantitative level in the cardiac muscle (rat right ventricle trabecula at 27 °C), through a spatial resolution of the small and ultra-small angle X-ray diffraction [19]. The analysis, besides supporting the mechanosensing hypothesis showing an increase in the axial periodicity of the filament backbone faster than the force development, suggested that in the relaxed cardiac muscle (diastolic phase), while all the motors are folded against the thick filament, only a sub-population in the C-zone lie in a helical track. This result was also supported from RLC-probes analysis in relaxed skinned trabecula of rat which indicated only about one-third of motors were in an IHM-like configuration [48]. This suggested a different stability of motors in different regions upon activation. Motors at the tip (D-zone) of the thick filaments activate first, generating a tension which is properly distributed along all the thick filament, and which helps to activate other motors. At the peak of the twitch, half of the helically ordered motors left this conformation, and only 10% of the motors, all confined in the C-zone, were attached, and generate the force [19], a value close to that obtained by a recent independent mechanical analysis [95]. This is coherent both with the lower tension present in the D-zone because force is developed from each motor toward the M-line, as was also proposed by a mathematical model [96], and with the MyBP-C sensitization of the thin filament to calcium [6,97].

The deactivation of the thick filament following a rapid shortening (20% L0) was also showed in activated demembranated trabeculae from rat heart with probes attached to the N- and C-lobes of the RLC [75]. Both the probes showed about 66% of recovery toward the relaxed value after only 30 ms. The extent of the recovery was proportional to the extent of the shortening, from 5% to 20%, and, in the case of the N-lobe, linearly proportional to the final force, while the analysis showed a degree of flexibility between the two lobes. When the shortening was imposed in ramps with different velocities or in subsequent steps, the probe on the N-lobe showed a remarkably linear relationship with the SL, with little sensitivity to actual force or imposed shortening velocity. The data suggested that the C-probe is associated with the attachment and detachment of the motors, while the N-probe is a mark of a structural transition linked to the regulatory system, but that overall, the orientation of the RLC, and thus the regulation of the thick filament, depends on the modifications of the SLs during contraction, in a time scale relevant for the single heartbeat.

Taking all this evidence together, after its postulation in amphibian skeletal muscle, the mechanosensing mechanism was also confirmed, with strikingly similarities, in mammalian skeletal and cardiac muscle, showing that the thick filament regulation, as the SXR state, is well preserved across species. However, fundamental differences also emerged. Applying a similar length-stretch protocol, described before for the passive skeletal muscle [43], to demembranated rat trabecula with RLC probes, at quasi-physiological temperatures [48], allowed a comparative analysis of the role of the active and passive tension on the thick filament activation in cardiac and skeletal muscle. The activation effect of the passive filament stress was almost the same in relaxed skeletal and heart muscle, but the low values reached in physiological conditions in the latter, below 20 pN per thick filament, affected only very slightly the motor conformation [48] (however, see also the open debate reported in the next section). Moreover, while in skeletal muscle, the passive and active tensions had similar effects, in cardiac muscle, the thick filament activation had an eight-fold higher sensitivity to active than passive stress, likely matching the physiological levels of the systolic tension [48].

The fine regulation of the OFF–ON equilibrium is important not only in the process of force generation, but also in the muscle relaxation, which is crucial in cardiac muscle for a correct ventricular refilling. The X-ray signatures of the thick filament activation were also used to follow its deactivation, with the high time and spatial resolutions described before, on the tension decay phase of the twitch in rat cardiac trabecula [19]. Muscle relaxation followed a precise scheme with three kinetic phases. From Tp to 220 ms after the stimulus, few motors detached, and the recovery of the OFF state was limited to the D-zone, where the active tension was lower because the Tp was supported by motors in the C-zone, as said before. Then, a second phase from 220 to 300 ms was characterized by a further detachment of motors with the switching OFF of the C-zone but without the recovery of the helical configuration. Only in the last phase, from 300 ms, the motors started to recover the helical configuration and reached fully the structural configuration characteristic of the diastolic phase, showing that the folding of the motors was kinetically dissociated from the packaging in the helical structure [19].

Similarly, in the single twitch analysis in mouse EDL at 28 °C, the packed configuration, and thus the helical order, was not recovered during the initial phase of relaxation, which was coherent with the high tension which characterized this phase [91]. However, even at the end of the mechanical relaxation, when the force dropped to less than 5% Tp, while X-ray signatures of the packed configuration reached the pre-stimulus levels, the helical order had an evident delay, and only 80% of the motors recovered it. An even bigger delay was observed in whole mouse soleus muscle at a lower temperature (22 °C) [98], where the spacing of the third order of meridional reflection after tetanic stimulus returned to the pre-activation levels after about one second. This effect may have a role in the post-tetanic potentiation, the higher twitch force observed after a tetanus, and represent an on-going field of study. It is important to note that soleus muscle is composed of a great percentage of slow-twitch fibers and that the observed differences may be associated with a difference in the thick filament activation and deactivation mechanism with respect to the fast-twitch muscles.

The proven ability of the thick filament to regulate the available motors in a very rapid way, sensing the force that is acting on it, the mechanosensing mechanism, is crucial in understanding the regulation of muscle contraction. However, in conclusion of this section, it is important to note that one or more complementary mechanisms may be associated with it. Notably, the higher resolution X-ray analysis made during an imposed unloaded shortening at maximum velocity on mouse EDL muscle, showed a small but reproducible modification of the thick filament structure [92], which was not detected before [18]. Considering the very low force acting in these conditions, an additional signaling pathway, besides the mechanosening mechanism, may then be present. A possible role can be played by the calcium itself. It was shown that the binding site for divalent cations on the RLC plays an important role in stabilizing the SRX [99]. The site primarily binds the Mg^2+^, but at high Ca^2+^ concentrations, there is a partial substitution, and the authors suggested that it can help in the thick filament regulation. Some findings support this view, but at very high Ca^2+^ concentrations [100]. A very recent report on a small molecule inhibitor of the thin filament may shed light on the role of Ca^2+^ in activating the thick filament. This inhibitor did not affect the actomyosin interaction, having no effect on the actin-S1 ATPase rate, but strongly inhibited the actin regulatory system, dropping the ATPase rate in a regulated thin filament (RTF)-S1 system and keeping the force to zero at activating calcium concentration in porcine myocardium [101]. Surprisingly, despite the force being absent, X-ray signatures showed an activation of the thick filament at several physiological pCa. Quantitatively, in those experimental conditions, the presence of the force activated the thick filament more, but most of the activation was supported by the calcium. High [Ca^2+^] also activated reconstituted cardiac synthetic thick filaments, which was shown to reproduce the SRX stability [52], but not in several constructs where the thick filaments cannot be fully formed [101]. However, as noted by the authors, these data were obtained in a steady state condition, and it is not known if the Ca^2+^-mediated activation is important in the single beat time scale. In skeletal muscle, it would be in contrast to the non-activation of the thick filament during the fast shortening imposed at the end of the latent period [18,92].

However, among all the pathways of regulation of the thick filament, the tension-mediated regulation, provided by the mechanosensing mechanism, appears to be the most readily available to the muscle in the time scale of a physiological contraction (single beat in cardiac muscle). Moreover, the two-way relationship shown, from the activation of the thick filament to the mechanics of the muscle, and from the mechanical perturbation to the thick filament activation, represents an important tool to understand the in vivo adaptation of muscle regulation through the mechanosensing mechanism.

## 7. Influence of Mechanical Modulation of the Thick Filament Activation on Contractile Performance

The first evidence that a tension-induced regulation of the thick filament can influence the subsequent contraction was already provided in the first work, which showed the mechanosensing mechanism itself [18]. Reducing from 10% to 5% L_0_ the extension of shortening superimposed to the fully activated, isometrically contracting frog fiber, as described above, the X-ray analysis showed that the thick filament switched OFF to a lower extent. The redevelopment of force after the two shortenings could be fitted by two exponentials characterized by different rates, the one with the smaller population of OFF motors being faster. Both rates were faster than the force development after the latent period or after the initial shortening imposed just after the latent period, which had similar kinetics. The interpretation proposed by Linari and coworkers was that the slower tension recovery was due to the time needed by the thick filament to reach, or recover, the fully ON state, depending on the degree of inactivation, with the rates starting from the fully OFF thick filaments being the slowest. This interpretation found support also from in silico analysis, in the first mathematical model which included the mechanosensing mechanism [102]. The analysis made on mammalian skeletal muscle highlighted a small discrepancy in the rate of force development with or without the imposed shortening after the latent period, the former being slightly faster than the latter (time to 0.5 T_0_ delayed by only 5.4 ms despite the delayed force rise of 8.8 ms) [92]. This could be related to the lower thick filament activation detected by the higher resolution at the end of the zero-load shortening, described in the previous section. Notably, despite being less easily interpreted because of the more complex system, a similar relationship was also shown in demembranated rat cardiac trabecula [75], with slower force recovery at higher imposed shortening, related to higher levels of deactivation of the thick filament. A coherent result was also reported in cardiac muscle carrying the R58Q mutation, associated with a severe HCM phenotype, which affects the thick filament regulation [103]. Signals from probed RLCs showed that the mutation stabilized the OFF conformation of the thick filament both in relaxed and activated conditions, leading to a lower maximum force. The stabilization of the OFF state was also confirmed through ATPase rates analyses. A release-stretch protocol showed a slower rate of force re-development in mutant trabecula than in WT trabecula.

The proven ability of muscles to regulate their contractile performance through a mechanically mediated regulation of the activation of the thick filament has a clear impact on physiological contractions, at least in cardiac muscle. As was already mentioned, through this ability, cardiac muscle rapidly adapts the number of active motors to the external conditions starting from the same diastolic situation (same initial SL) [94]. Instead, some contradictory results are present about the role of the mechanosensing in the regulation of the diastolic level of activation of the thick filament at different SLs, and its influence on the Length-Dependent Activation (LDA). LDA is usually tested comparing force-pCa curves at different relaxed SLs and can be seen as the cellular counterpart of the fundamental auto-regulatory property of the heart known as Frank Starling relation. LDA is characterized by both higher tensions at maximally activating Ca^2+^ and higher calcium sensitivity at longer SLs. Following the mechanosensing paradigm, the effect of the diastolic SL on the thick filament activation can be mediated by titin through higher passive forces at higher initial SLs [104]. Genetically modified rodents with more compliant titin showed reduced LDA [105,106]. Very recently, to avoid secondary effects usually linked to fibers carrying mutated titin, Hessel and co-workers proposed an engineered mouse model with a cleavage site in the titin, allowing the analysis of the reduction in the titin force on the same sample, before and after a 50% titin cleavage [107]. In these permeabilized fiber bundles at 24 °C, the titin cleavage reduced the passive force at any SLs, as well as the X-ray signature of the activation of the thick filament. LDA was not completely prevented, but coherently, the calcium sensitivity was reduced in the post-treatment fibers.

However, differently than in skeletal muscle, relaxed cardiac muscle showed conflicting results about a direct regulation of the thick filament structure through the SL variations. Evidence of a higher diastolic activation level of the thick filament at higher SLs was shown in demembranated cardiac myocytes with the RLC probes [59]. X-ray diffractions on WT and mutant rat trabecula, associated with a lower passive force, showed a correlation of the LDA and a titin-mediated structural rearrangements in the thick filament in the relaxed state, despite X-ray signals not directly indicating a higher activation at higher SLs in diastole [106]. Instead, in permeabilized porcine myocardium at 22 °C, increasing the SLs from 2.0 µm to 2.3 µm under relaxing condition moved both the X-ray signatures of the OFF state of the thick filament and the mant-ATP turnover measuring the amount of SRX population, toward a higher population of ON motors [82]. However, X-ray diffraction patterns in diastolic intact trabeculae from the rat heart were not affected increasing SLs, despite the fact they doubled the peak force in the subsequent twitch [19,79]. Contrary to demembranated trabecula then, the results on intact preparations seem to be against the idea that LDA is mediated by a thick filament regulation triggered by the passive tension generated by titin. Besides a possible role of the different relative composition of fast- and slow-twitch fibers in rodents and large mammalian heart muscles, a suggested explanation for the different results is that the protein–protein interactions which stabilize the OFF state are weakened in permeabilized myocytes [79]. As an alternative explanation, it may be possible that in the intact preparation the weakened interactions produced by higher SLs can be undetectable [82]. In this respect, it is worth noting that the first mathematical model which included the mechanosensing mechanism in cardiac muscle, suggested a titin contribution to the LDA not through a direct modification of the thick filament structure in diastole, but through the contribution of the passive force during the subsequent phases of the twitch [108]. The passive tension in the model was, in fact, lower than the imposed threshold in the thick filament activation, suggested to be about 10% T_0_ for skeletal muscle [18]. The experimental data described above still seems to be coherent with the proposed mechanism. Overall, the existence of a threshold in the mechanosensing mechanism, and its perturbations, may play an important role in muscle fine regulation, as also suggested by the passive length-stretch protocol applied to demembranated rat trabecula with probes on the RLC, at quasi-physiological conditions [48]. At low [Ca^2+^] (both pCa9 and the more physiological pCa7), the stretches of the SL in the range of 2.0 to 2.3 µm had almost no effects on the folded conformations of the myosin motors, even at physiological levels of phosphorylation. However, at pCa 6.6, which slightly activated the thin filament but only up to 3 kPa, increasing the SL showed a clear increase in the number of the myosin available to work, and then of the total tension, a well-known phenomenon called stretch activation (SA). Importantly, these changes in the RLC orientation were similar even if the activation followed the stretch, which is the physiological sequence of the LDA, suggesting that SA and LDA share the same mechanism, linked to the thick filament activation. This result is even more relevant for the understanding of the LDA, considering that in intact cardiac tissue, at least in some pathological conditions, diastolic stiffness has a significant contribution from actomyosin-related force [109]. Coherently, an inhibitory effect of mavacamten was reported on cardiac passive force in human multicellular preparations at 37 °C [57] and human engineered heart tissue [81]. Moreover, in view of the mechanosensing mechanism, it is crucial to distinguish between SL and passive force. In intact mice cardiomyocyte with a more compliant titin isoform than in WT, while the systolic force-SL relation was blunted as expected, the relation between systolic and end diastolic forces was not significantly different [110]. This result indicated that LDA is not determined by the SL in itself, but rather by the passive force generated at a given SL, possibly because of the role of the mechanosensing mechanism.

Apart from the diastolic conditions, the role of the thick filament regulation in the LDA is supported by several recent data, even though the complexity of the system makes it difficult to highlight its role uniquely. To this aim, the elongation of the SL has been studied in association with different perturbations of the OFF–ON equilibrium. Strong evidence came from probes which reported structural changes in both thick and thin filaments of rat cardiac muscle at different calcium concentration and two SLs, 1.9 µm and 2.3 µm [111]. The analysis suggested that the thin filament activation was not affected by SL at maximum calcium activation, possibly because it was already fully activated at shorter SLs. However, at longer SLs, the thin filament showed a higher calcium sensitivity, notably even after complete force inhibition by blebbistatin. The thick filament, instead, was more activated both in low and, with a higher extent, high pCa [111]. The same study extended the analysis using probed RLCs containing the R58Q mutation, which, as said, stabilizes the OFF conformation of the thick filament. The force-pCa curve at 1.9 µm and 2.2 µm in WT and mutant trabeculae showed that, in the latter, both the maximum tension and the structural modifications of the thick filament were almost independent of SL, while the increase in the calcium sensitivity was preserved [103]. Therefore, the activation of the thick filament seems to have a role in the higher tension shown by LDA [103,111]. This result is also supported by in silico analysis [96,112].

More recently, X-ray and biochemical studies on permeabilized porcine fibers in relaxing conditions at 22 °C proved that LDA involves the recruitment of motors from the SRX state, associating it with the treatment with mavacamten [82]. Physiologically relevant concentrations of mavacamten restored almost all SL-induced thick filament structural alterations, despite LDA still being present. Mavacamten reduced maximum tension at both SLs and all [Ca^2+^], as expected, from its stabilizing effect of the SRX. However, while its ability to reduce maximum tension was the same at both SLs, mavacamten abolished the variation of pCa50 with the stretch. The persistence of LDA in the presence of mavacamten was confirmed also in human engineered heart tissues at 36 °C undergoing constant slow linear stretch while contracting at 1 Hz [81], and on force-pCa curves of multicellular preparations from human donors at 37 °C [57]. In the former work, 0.33 µM of mavacamten treatment also increased the gain of active force after stretch, when it was normalized to the initial force level, which was lower in the presence of the compound. However, this enhanced Frank–Starling response was not observed at a higher concentration of mavacamten or in the latter work, at least at statistical significance, despite there being a trend toward a higher increase in pCa variations.

Altogether, the reported data convincingly suggest that the mechanosensing mechanism has a physiological role in the fine muscle regulation. However, dissecting it in the complex muscle regulatory system requires more data, as shown by the increasing number of studies in the literature with this aim.

## 8. Proposed Working Model

A single, comprehensive, mechanistic interpretation of thick filament regulation, and its function on the pathophysiology of muscle performance, is still missing. A limit is due to the different conditions used in the experiments, as emerged in previous sections. The temperature and the lattice spacing in the demembranated preparations play an important role in the stability of the OFF motors, and differences in these values may blunt, or emphasize, the role of the thick filament regulation. Additionally, several experimental techniques available to test its mechanical implications require demembranated muscle fibers, a preparation which seems to weaken the stability of the OFF state, possibly reducing the activating threshold. Different techniques probe the stability of different “OFF” states (IHM, SRX, folded motors with or without helical periodicity), whose relationship is not yet fully understood. Moreover, despite the reported evidence that the mechanical inactivation of the thick filament has been conserved during evolution, and that its regulation is preserved in different species and muscle types, important differences are present, not only between smooth and striated muscle, but also between cardiac, fast and slow skeletal isoforms. These differences, such as the sensitivity to passive or active tension, and the population of OFF motors in the relaxed muscle or at its maximal physiological tension, are likely aimed to optimize the thick filament-based force regulation in relation to their different tasks.

Nonetheless, several works contributed to the definition of a “mechanosensing model”, where the thick filament activation is regulated by the tension acting on it, leading to an internal feedback system (see Figure 1).

The first idea of a dual filament regulation, in which both the thin and thick filaments can be active or inactive, was proposed by Linari and collaborators [18]. The thick filament has few constitutively ON motors which drive the unloaded shortening when the thin filament is activated by calcium. If the shortening is prevented, the tension generated by the constitutively ON motors drives the activation of other motors. This is also true when a passive tension is acting on the thick filament [43], suggesting a role of the SL before and during activation. The protein–protein interactions, which stabilize the OFF state, was postulated to stabilize also the shorter 14.34 nm periodicity of the thick filament, which reverts to the longer 14.57 nm periodicity when the packing is lost [18]. The stress-dependent activation implies a positive feedback loop where more attached motors accelerate the activation of the still inactive motors during the force generation. A finer regulation can be reached through different degrees of stability along the thick filament. SRX motors can be in a folded helical and a folded non-helical configuration, the former being concentrated in the C-zone, stabilized by the MyBP-C. The folded non-helical motors, along with the non-folded motors, are concentrated in the D-zone and toward the tip of the thick filament. This region is then the first to be activated after the stimulus, subsequently driving the activation of the whole filament, and the first to be deactivated, leaving the motors in the C-zone the only to sustain the peak of force during a twitch, at least in cardiac muscle [19]. The different configurations are linked to the flexibility between the C- and N-lobes of the RLC, the N-lobe being in an IHM-like configuration for both the populations, whereas only the folded helical motors also have the C-lobe stabilized in that conformation [48]. Moreover, the C- and N-lobes also have different roles during relaxation, which helps to explain the shortening-induced deactivation [75]. The blocked head may stay folded to the thick filament, while the front head attaches to actin and generates the power stroke. The power stroke itself would put the front head N-lobe in a conformation in which it can make the IHM-like intra-molecular interactions with its partner blocked head. These interactions could then catalyze the formation of the full IHM conformation once the front head detaches from the actin filament [75]. Finally, the different levels of stability of the auto-inhibited state of the myosin motors in different zones of the thick filament further support the evidence of a crucial role of the MyBP-C in the dual filament regulation of muscle contraction. Different domains of the MyBP-C can bound both the blocked and free heads in the dephosphorylated state through interactions with the RLC–neck region, while the MyBP-C N-terminal (C0) domain can randomly attach and detach from actin [64]. Phosphorylation of MyBP-C releases parts of it and, consequently, the associated heads, stabilizing the actin–C0 interaction. At activating calcium levels, another MyBP-C domain (C1) cooperatively interacts with the thin filament, fully activating it, but possibly also bringing the MyBP-C–attached myosin heads closer to the thin filament.

Besides this working model, it is important to remember that the activation process is a sequence of different steps, not all sensible to the tension acting on the myosin backbone. The single S1 can transiently assume an SRX conformation without directly forming the folded state, and this conformation is further stabilized by intra-molecular interactions [41]. The existence of an SRX state in the HMM can be interpreted as a folded configuration in the HMM [40] which is further stabilized by its folding back against the thick filament. Eventually, the heads open while the S2 is on the thick filament, or they move away from the thick filament while still interacting each other in a SRX state. In this conformation, the two heads either can open and become available to the actomyosin interaction, or the free head can weakly interact with actin to disrupt the IHM state [40].

## 9. Conclusions

The fine regulation of the thick filament activation is crucial in muscle performance, and its alterations lead to cardiac diseases with a strong impact on the society, in terms of both life and social costs. Within the complexity of the thick filament activation processes, the mechanosensing mechanism has a prominent role because it implies a feedback system which is readily available to the muscle to immediately adapt its functions to changing external conditions. Therefore, the understanding of its molecular basis is crucial in advancing our knowledge of muscle contraction and to design therapeutic interventions.

Models based on only two states of activation of the detached myosin motors, ON–OFF or SRX–DRX, and a switch based only on the tension acting on the thick filament, despite being informative, are too oversimplified to advance our understanding on the physiological meaning of this mechanism. The activation of the motors appears to be a multi-step process, driven by thermal fluctuations to overcome potential barriers, representative of several protein–protein interactions, which are regulated, or modified, by the tension, small molecules, genetic alterations as well as temperature, ionic strength, lattice spacing and so on. Being in equilibrium, even acting on only one of these stable states alters the population of motors in all the other states, with potential effects on the active force.

If the molecular bases of the mechanosensing mechanism, as proposed, are related to the force-induced perturbation of these molecular interactions, it may affect primarily the interaction between neighbor dimers, but less so the head–head interactions shown in the IHM conformation of the HMM, especially if the inhibited HMM moves toward the actin filament before being activated. Similarly, mavacamten may have a role in stabilizing the IHM in the whole thick filament primarily stabilizing the inhibited state in the HMM or even in the sS1, which would indirectly stabilize the whole thick filament structure. Moreover, in this scheme, the state occupied by one head influences the stability of the partner head and of the neighbor dimers, as also supported by the high cooperativity of the thick filament activation, which is higher than that of the force.

The effects on the threshold of activation, or on the number of constitutively ON motors, may also have a different role than the modification of the rate constants between states. These events must then be inserted into the holistic view of the dual filament regulation where the positive feedback loop drives both the thin and the thick filaments activations. All these aspects can be clarified, from the single molecule to the whole muscle, only through a multi-scale and multi-disciplinary approach: in vitro, in situ, in vivo and in silico.

## Figures and Tables

**Figure 1 ijms-24-06265-f001:**
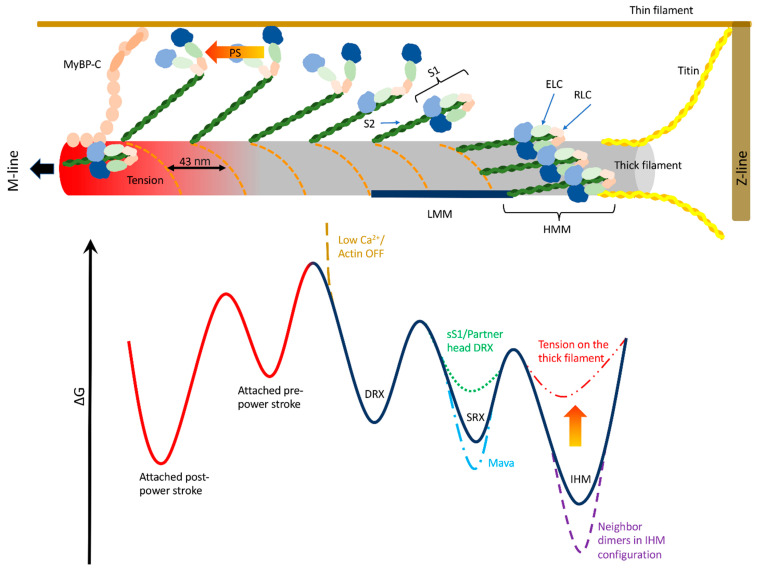
Schematic drawing of the myosin domains, of the thick filament activation (**upper** part) and of the myosin stable states energies (**lower** part). From top to bottom and from right to left: Thin filament, whose regulatory proteins are not shown, is activated by Ca^2+^ ions and is attached to the Z-line. Myosin LMM domains form the thick filament, attached to the Z-line via titin, while contiguous HMMs form the interacting head motif (IHM), stabilized by intra-molecular interactions, and helically folded motors, stabilized by inter-molecular interactions. Titin runs from Z-line to the M-line along the thick filament, not depicted for clarity. MyBP-C runs from thick filament, stabilizing its OFF state, to thin filament, sensitizing it to Ca^2+^. The single sS1 is formed by the motor domain the RLC and the ELC, and can either be in an autoinhibited super-relaxed state (SRX) or in a disordered relaxed state (DRX), pointing toward the actin filament and ready to create the actomyosin complex and generate force through the power stroke (PS). Tension (red) is transmitted along the thick filament from the attached motor toward the M-line. The bidimensional simplification of the energetic landscape of the myosin stable states (lower part of the figure) depicts the possible effects described in the text. The tension (red) destabilizes the helically ordered IHMs which are instead stabilized by the inter-molecular interactions through the neighbor dimers (purple). The autoinhibited state of the sS1 is stabilized by mavacamten (light blue) and destabilized by the loss of the intra-molecular interactions (green). When actin is non activated, at low calcium, the actomyosin formation is prevented (light brown energy barrier).

**Figure 2 ijms-24-06265-f002:**
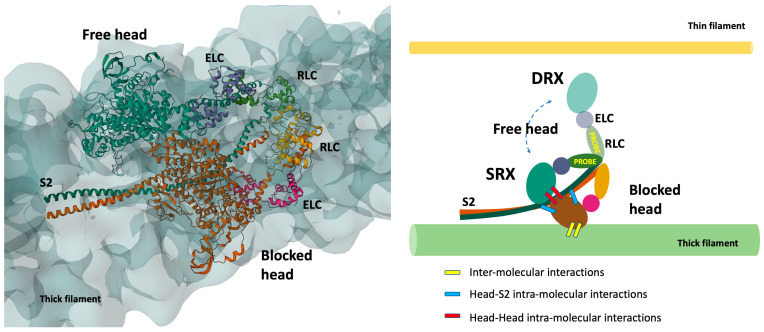
Schematic model of the Interacting Heads Motif (IHM), (**left** panel, based on PDB: 5TBY) and cartoon model of myosin interactions involved in the formation and stabilization of the IHM (**right** panel, based, redrawn and adapted from ref. [27]. Copyright 2017 E-life). IHM is associated with the SRX state, lying on the surface of the thick filament, with the blocked head primed in the pre-power stroke configuration, docked by intra-molecular interaction with its S2 (blue connections), and anchored by inter-molecular interactions with the neighboring myosin’s tail (yellow connections). The IHM is stabilized by intra-molecular interactions between the free head and the blocked head (red connections). The probes on the RLC are also schematically shown to describe the change in orientation during activation.

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
