# Peer review of "Muscle Mechanics and Thick Filament Activation: An Emerging Two-Way Interaction for the Vertebrate Striated Muscle Fine Regulation"

_ijms, 2023, doi:10.3390/ijms24076265_

Round 1

Reviewer 1 Report

Overall, this is a well-written review of thick filament activation in vertebrate muscle. I have one concern about the proposed models section. Mechanosensing model has been extensively studies bu different groups. But the way the section was constructed is too fragmented. Linari, Fusi, Brunello, Park-Holohan, Kampourakis, and Irving et al. are essentially shaping their models under the umbrella of the mechanosensing model. I would suggest the author write one model and cite the relevant publications. I don’t think Alamo, Rohde, and Anderson et al. proposed a convincing thick filament activation model in vertebrate muscle. They studied the structural basis of IHM and/or SRX but did not mention how they are being turned on. The only other preliminary model is Ma et al. in porcine myocardium proposed that both the passive stretch during diastole and calcium release during systole modulate thick filament activation.

   The rest are minor things:

1.      Line 26. There are considerable structural differences among striated muscle sarcomeres. Especially among different species. It might be worth throwing “Vertebrate” in the title and making it clear early in the introduction that this review is vertebrate muscle specific

2.      Line 46 “signalling” should be “signaling”

3.      Line 8 and line 48: the site that hid should be the myosin binding sites on thin filament, not acting binding sites

4.      Line 113. The authors mentioned two reflections from the helical order of the myosin heads, I assume they are myosin layer lines. Short axial periodicity I suppose, is M3 reflection. The 1% increase might be SM3 increases 1%. Please clarify it. Two signatures should be losing of helical ordering of myosin heads as evidenced by the decrease of MLLs and the increase of myosin axial periodicity as evidenced by the increase of SM3. While the loss of MLLs intensities are not controversial, whether the increase in M3 spacing is the cause or the effects is debatable. I would suggest to the author adding something like “as proposed in mechanosensing mechanism by Linari et al.

5.      Line 130 to 132: The meanings of inter and intra are not clear. The canonical IHM has the heads folded on its tail, so it is not inter-molecular interaction. Head-head and head-tail interactions form IHM.

6.      Line 134: Another thing worth noting is that the tarantula thick filament has four-fold symmetry. It is unclear whether the 3-fold symmetric vertebrate muscle can form interactions between the neighboring heads.

7.      Line 486. Another thing also worth noting is that this study and the following studies from the Florence and KCL groups are all from fast-twitch muscles. Studies from mouse Soleus muscle (mixture of fast and slow) and rat soleus muscle showed that mechanosensing might work differently between different muscle types.

8.      Line 627. It is worth noting that soleus is composed of 40% of sow fibers so the longer delay is not simply a temperature effect.

9.      Line 735. The author should also consider the difference between rodent and large mammalian hearts, in which thick filaments are composed of different myosin isoforms.

Reviewer 2 Report

The state-of-the-art of reciprocal interactions between activation of the thick (myosin) filaments and mechanics of striated muscles is reviewed. Although mechanical activation of thick myosin filaments was discovered only 8 years ago, several reviews of this phenomenon have been published since. Nevertheless, new data are accumulated quickly, so reviewing recent findings and their relations to previous works is useful and timely.

I have several comments and suggestions which I hope can help the review to be more useful and easier to perceive for readers not deeply involved in the problem.

1.  The review is missing a detailed definition of IHM, SRX, and their interrelation with the open and closed states, steps of the ATP hydrolysis cycle for myosin head, and a description of experimental methods used for their detection. The helical arrangement of myosin heads on the surface of the thick filament detected by x-ray diffraction requires interaction between neighbor myosin molecules not only the interaction of two heads of a molecule that reduces their ATPase activity. A discussion of the interplay between the intra- and intermolecular interactions and their role in myosin regulation is also missing.

2.  IHM was discovered by Wendt e.a. [Wendt T, Taylor D, Trybus KM, Taylor K. Three-dimensional image reconstruction of dephosphorylated smooth muscle heavy meromyosin reveals asymmetry in the interaction between myosin heads and placement of subfragment 2. Proc Natl Acad Sci U S A. 2001; 98(8):4361-6. doi: 10.1073/pnas.071051098] and then found in other muscles. I think the paper should be cited in the review.

3.  A picture with a detailed IHM structure showing the difference between the blocked and free heads and demonstrating the details of the head-head and head-S2 interactions, together with the localization of the phosphorylation sites and the positions of fluorescent probes on RLC, would be very useful.

4.  Lines 109-123. In a review of the x-ray diffraction evidence for the loss of the helical order of myosin heads upon activation, along with early works of the 1970-s [20-22], I would recommend citing the first time-resolved synchrotron experiments which have demonstrated the time course of tension development and myosin helix disordering in twitches and short tetanus [Huxley HE, Faruqi AR, Kress M, Bordas J, Koch MH. Time-resolved X-ray diffraction studies of the myosin layer-line reflections during muscle contraction. J Mol Biol. 1982; 158(4):637-84. doi: 10.1016/0022-2836(82)90253-4].

5.  Lines 139-141. The presence of IHM in all types of muscles was challenged by Knupp C, Morris E, and Squire JM [The Interacting Head Motif Structure Does Not Explain the X-Ray Diffraction Patterns in Relaxed Vertebrate (Bony Fish) Skeletal Muscle and Insect (Lethocerus) Flight Muscle. Biology (Basel). 2019; 8(3):67. doi: 10.3390/biology8030067] who claimed that the structure is absent in relaxed fish muscles with simple crystalline myofilament structure. However, later this statement have been refuted by x-ray ray diffraction experiments analyzed using cryoEM data and direct modeling [Koubassova NA, Tsaturyan AK, Bershitsky SY, Ferenczi MA, Padrón R, Craig R. Interacting-heads motif explains the X-ray diffraction pattern of relaxed vertebrate skeletal muscle. Biophys J. 2022; 121(8):1354-1366. doi: 10.1016/j.bpj.2022.03.023]. Mentioning these recent publications seems to be appropriate in the context of the review.

Another paper that can be mentioned in the discussion of the validity and consistency of the x-ray and cryoEM data in studying myosin regulation is [Padrón R, Ma W, Duno-Miranda S, Koubassova N, Lee KH, Pinto A, Alamo L, Bolaños P, Tsaturyan A, Irving T, Craig R. The myosin interacting-heads motif present in live tarantula muscle explains tetanic and posttetanic phosphorylation mechanisms. Proc Natl Acad Sci U S A. 2020; 117(22):11865-11874. doi: 10.1073/pnas.1921312117] where cryoEM data were used for calculating the x-ray diffraction pattern from life tarantula muscle and revealed good quantitative agreement of these two approaches.

6.  Lines 180-183. It is worth mentioning that stretch-dependent changes in the orientation of the fluorescent probes on myosin RLCs were observed in [Ref 37] at sarcomere length far beyond its physiological range.

7.  Lines 194-194. An increase of the helical order of myosin heads with temperature in mammalian skeletal muscles was discovered by J Wray [Wray, J. 1987. Structure of relaxed myosin filaments in relation to nucleotide state in vertebrate skeletal muscle. J. Muscle Res. Cell Motil. 8:62a. (Abstr.)] and later studied by many research groups, for example [Lowy J, Popp D, Stewart AA. X-ray studies of order-disorder transitions in the myosin heads of skinned rabbit psoas muscles. Biophys J. 1991 Oct;60(4):812-24. doi: 10.1016/S0006-3495(91)82116-6; Malinchik S, Xu S, Yu LC. Temperature-induced structural changes in the myosin thick filament of skinned rabbit psoas muscle. Biophys J. 1997 Nov;73(5):2304-12. doi: 10.1016/S0006-3495(97)78262-6].

8.  Lines 268-274. The already mentioned paper by Padron et al., (PNAS, 2020) shows an interaction between RLC phosphorylation, IHM formation, and mechanical performance in tarantula muscles.

9.  The last parts of the review (sections 6 and 7) need to be more structured and logical instead of listing different works with varying degrees of detail.
